# The Validation of a Precursor Lesion of Epithelial Ovarian Cancer in *Fancd2*-KO Mice

**DOI:** 10.3390/cancers15092595

**Published:** 2023-05-03

**Authors:** Sarah Sczelecki, Janet L. Pitman

**Affiliations:** The School of Biological Sciences, Te Herenga Waka Victoria University of Wellington, Wellington 6012, New Zealand

**Keywords:** epithelial ovarian cancer, precursor lesion, mouse model

## Abstract

**Simple Summary:**

The advancement of improved early detection is in part hampered by limited knowledge of ovarian cancer (OC) origin, difficulties in accessing early-stage OC in female patients and a lack of animal models to study early OC development. The *Fancd2* knock-out (KO) animal model has formerly been shown to develop epithelial ovarian cancer, with the purported precursor lesion hypothesised to develop from primordial ovarian follicles prematurely devoid of their germ cell. This hypothesis was based on our extensive knowledge of follicle development and qualitative observation. Therefore, for the first time, we sought to demonstrate a relationship between the proposed precursor cells and a late-stage tumour using gene expression analysis, providing genetic evidence to support the use of this animal model as a novel tool to study early epithelial ovarian cancer.

**Abstract:**

Ovarian cancer (OC) has the highest mortality rate of all gynaecological malignancies. The asymptomatic nature and limited understanding of early disease hamper research into early-stage OC. Therefore, there is an urgent need for models of early-stage OC to be characterised to improve the understanding of early neoplastic transformations. This study sought to validate a unique mouse model for early OC development. The homozygous Fanconi anaemia complementation group D2 knock-out mice (*Fancd2^−/−^*) develop multiple ovarian tumour phenotypes in a sequential manner as they age. Using immunohistochemistry, our group previously identified purported initiating precursor cells, termed ‘sex cords’, that are hypothesised to progress into epithelial OC in this model. To validate this hypothesis, the sex cords, tubulostromal adenomas and equivalent controls were isolated using laser capture microdissection for downstream multiplexed gene expression analyses using the Genome Lab GeXP Genetic Analysis System. Principal component analysis and unbiased hierarchical clustering of the resultant expression data from approximately 90 OC-related genes determined that cells from the sex cords and late-stage tumours clustered together, confirming the identity of the precursor lesion in this model. This study, therefore, provides a novel model for the investigation of initiating neoplastic events that can accelerate progress in understanding early OC.

## 1. Introduction

Ovarian cancer (OC) is the eighth most diagnosed cancer and the fifth leading cause of mortality in women. Around 90% of OC diagnoses are malignant OC of epithelial origin (EOC) [1,2]. Poor survival rates are perpetuated by its asymptomatic nature and lack of early detection methods [3]. Research into the early detection of EOC is hampered by divergent views on the origin of EOC and, therefore, the identity of the initiating precursor cells. Historically, EOC was thought to be solely derived from the ovarian surface epithelium (OSE) [4], based not only on its transformation into EOC in vitro [5,6,7,8] and in vivo [9,10,11], but also on evidence of its involvement in the incessant ovulation hypothesis [12]. This hypothesis is based on the relationship between reduced numbers of ovulations and a subsequent decrease in the frequency of OSE damage and repair. It is further evidenced by a reduction in OC risk in multiparous women and those taking long-term contraception [13].

However, an alternative theory emerged where EOC was suggested to be of extra-ovarian origin, derived instead from the fallopian tube epithelium (FTE). According to this theory, tumour formation occurs via the deposition of malignant cells onto the ipsilateral ovary and invagination occurs via inclusion cysts [14]. This theory is based on histopathological evidence, in particular, the presence of hyperplastic lesions (i.e., serous tubal intraepithelial carcinomas, STICs) on the fallopian fimbriae of women with *BRCA* mutations and the ability of tubal secretory cells to transform into OC in vitro [15]. This alternative theory also sought to reconcile incongruencies between the extra-ovarian histology of certain ovarian cancer subtypes (i.e., serous, endometrioid, clear cell or mucinous) and the lack of similar cell types found within normal intra-ovarian tissue. However, the origin of EOC is not mutually exclusive, and EOC developing from either OSE or FTE have unique characteristics, including latency, transcriptome and metastatic behaviour, further adding to the complexity of the disease [16].

We recently reported a new paradigm for the origin of EOC [17] developed from our extensive research on folliculogenesis in the developing sheep ovary [18] and knock-out (KO) mouse models [17]. In the Fanconi complementation group D2-KO (*Fancd2^−/−^*) mouse model, premature oocyte loss occurs due to impaired DNA repair and meiotic maturation, followed by the systematic development of OC [19]. Importantly, these mice do not undergo ovulation or display any morphological signs of tumorigenesis in adjacent tissues up to one year of age [17]. Central to this paradigm is that the pregranulosa cells in primordial follicles are held in a multipotent state that is under strict regulatory control by their associated germ cell (i.e., a naïve state) [20,21]. Upon activation of growth, the oocyte secretes a milieu of growth factors that direct these associated naïve cells down a differentiated granulosa cell pathway [22,23,24]. However, if oocytes are lost prematurely from primordial follicles, these naïve cells lose their germ cell regulator and undergo a series of abnormal pathological events leading to the development of heterogeneous ovarian tumours. These histological transformations are predictable with age and include the sequential development of sex cords, intraovarian epithelial ducts, OSE invaginations, tubulostromal adenomas, luteomas and cystadenomas of papillary or mucinous phenotype and finally, the integration of multiple tumour phenotypes including adenocarcinomas exhibiting an invasion into extraovarian tissues [17].

Although these three origin stories appear disparate, the common developmental origin of the OSE, FTE and granulosa cells (GC) is often overlooked. Specifically, the coelomic epithelium gives rise to the OSE and is adjacent and contiguous with the region that forms the Müllerian ducts, which eventuates into the epithelia of the oviduct, endometrium, cervix and distal fimbriae [1]. Additionally, the OSE is the primary source of GCs in ovarian follicles of sheep [18], mice [25], primates [26] and humans [26], providing a route for multi-potent cells to enter the ovarian cortex. Therefore, these multipotent pregranulosa cells, without their germ cell regulator, can retain an epithelial cell lineage and thus can subsequently undergo neoplastic transformation into the heterogenous EOC phenotypes, similar to what is observed in OSE- and FTE-derived OC.

Regardless of the origin of EOC, there are few investigations that have explored the initiating events and precursor cell types involved in EOC tumorigenesis. It is well accepted that high-grade serous OC derived from the FTE has an associated precursor lesion, namely STICs [27,28,29,30], but tumorigenesis does not exclusively occur through this initial phenotype [30]. In contrast, EOC originating from OSE is less understood; however, it is known to advance through the development of OSE into cortical inclusion cysts that undergo neoplastic transformations within the ovary proper [1,31]. Whilst the *Fancd2^−/−^* model has been used to study the spontaneous development of epithelial cancers [32,33] and a proposed precursor lesion (i.e., the sex cord) has been identified [17], this model remains genetically uncharacterised.

In this study, we examined the expression of EOC-associated genes in sex cords and a late-stage OC phenotype in *Fancd2^−/−^* mice, alongside normal ovarian tissue in control mice. We show for the first time that the gene expression signature of the precursor lesion (sex cords) identified in young *Fancd2^−/−^* mice [17] has significant similarities to a late-stage (adenoma) phenotype, including an increase in the expression of key epithelial markers. This study validates the *Fancd2^−/−^* model for studying early tumorigenesis of EOC and genetically links these phenotypes. 

## 2. Materials and Methods

### 2.1. Mice

*The Fancd2^−/−^* and *Fancd2^+/+^
*(wild-type) mice used in this study were from the breeding colony established at Te Herenga Waka Victoria University of Wellington (VUW) following the kind donation of mice from Professor Markus Grompe (Oregon Health and Science University, Portland, OR, USA) [32]. As *Fancd2^−/−^* mice are infertile, heterozygous *Fancd2^+/−^* mice were bred to obtain the *Fancd2^−/−^* experimental animals and their littermate wild-type controls. Genotyping was performed using end-point PCR as described in [32]. Mice were maintained on a 129S4 background, and all animal experiments were performed in accordance with the VUW Animal Ethics Committee pursuant to the Animal Welfare Act of New Zealand. Animals were maintained on a 12 h light–dark cycle and standard chow diet ad libitum.

### 2.2. Laser Capture Microdissection

Trimmed ovary pairs were collected from 3- and 12-month-old *Fancd2^+/+^* and *Fancd2^−/−^* mice (*n* = 4–6 per group) and flash frozen in optimal cutting temperature medium (Sakura Finetek, Torrence, CA, USA) within cryomolds on dry ice and stored at −80 °C until required. Laser capture microdissection (LCM) of specific cells was performed using a Molecular Machines & Industries (MMI) CellCut Laser Capture System with a UV laser (MMI GmbH, Eching, Germany), and the microdissected areas were transferred onto specialised caps of 0.5 mL diffusion tubes (MMI GmbH, Eching, Germany). The areas selected for microdissection were sex cords and a late-stage tumour phenotype from 3-month and 1-year-old *Fancd2^−/−^* mice, respectively (Figure 1). The microdissected control tissues included naïve GCs from primordial and primary follicles and differentiated GCs from growing follicles of *Fancd2^+/+^* mice (Figure 1). Since the *Fancd2* model is a global knock-out, to ascertain whether gene expression differences were directly related to the phenotype, rather than indirect differences as a result of the *Fancd2* knock-out, ovarian stromal tissue was collected from 3-month *Fancd2^+/+^* and *Fancd2^−/−^* mice. Only three-month-old animals were used as stromal control samples, since discernible stromal cells separate from tumour tissue were not identifiable in 1-year *Fancd2^−/−^* animals, preventing appropriate comparison to the 1-year *Fancd2^+/+^* ovarian stroma. After all the sections in each slide were collected, an aliquot of 100 µL of QIAzol lysis reagent (Qiagen, Germantown, MD, USA) was added to each cap, vortexed for 1 min and stored at −80 °C until required. A detailed LCM protocol is found in the Appendix A.

### 2.3. RNA Extraction

The microdissected tissue samples from the same biological replicate were pooled, and RNA was extracted using the miRNeasy Micro kit (Qiagen, Germantown, MD, USA) and treated with TURBO™ DNase (Ambion, Life Technologies, Vilnius, Lithuania), as per manufacturer’s instructions. Quality control of extracted RNA was performed using the Qubit RNA HS Assay kit (Life Technologies, Vilnius, Lithuania) and the Bioanalyzer Agilent RNA 6000 Pico Kit (Agilent Technologies, Victoria, Australia). Samples that were within the detectable limits of these quantification technologies all exhibited a RIN ≥ 6.

### 2.4. Multiplex GeXP Assay

Three sets of ~30 gene targets each were selected based on the current literature and various sequencing studies of high-grade serous ovarian carcinomas (Appendix A–S3). NCBI BLAST was used to design suitable gene-specific primers for reverse transcription and PCR amplification. Each primer set was designed to generate differentially sized amplicons, separated in theoretical size by at least five base pairs, in each multiplex gene set. For those samples where the level of RNA was quantifiable, 10 ng of RNA per multiplex gene set was used as a template for the reverse transcription (RT). For those samples where the RNA was below the detection limit of the Qubit and Bioanalyzer, the entire RNA sample was split into three equal volumes for each GeXP gene set. All experiments included no template and no enzyme RT negative controls. The resultant cDNA was halved, and each PCR reaction was performed as two technical replicates (as per manufacturer’s instructions, Beckman Coulter, Brea, CA, USA). The resultant PCR products were separated using gel capillary electrophoresis on the Genome Lab GeXP machine, followed by relative gene expression analysis. A detailed GeXP Assay protocol is found in the Appendix A.

The three gene sets included markers for folliculogenesis, cell cycle regulation, apoptosis DNA repair, tumour markers and suppressors; many of which are dysregulated in EOC (Appendix A).

### 2.5. Immunohistochemistry

For OC biomarker analyses, the fresh frozen cryopreserved 1-year *Fancd2^−/−^* samples that were used for LCM were assessed with IHC. The cryo-blocks containing the remaining tissue samples frozen in optimal cutting temperature medium were removed from −80 °C and immediately submerged in ice-cold 4% (*w*/*v*) paraformaldehyde for 1 h. To remove residual dissolved optimal cutting temperature medium, samples were then placed in fresh 4% (*w*/*v*) paraformaldehyde to continue fixation overnight at 4 °C and stored in 70% (*v*/*v*) ethanol until the tissue embedding step. Due to limitations in tissue size, samples from 3-month *Fancd2^+/+^* and *Fancd2^−/−^* mice were wholly used for LCM; therefore, additional samples from 3-month mice were used for IHC. These samples were freshly dissected and immediately fixed in 4% (*w*/*v*) paraformaldehyde overnight at 4 °C and stored in 70% (*v*/*v*) ethanol until required. All samples were then paraffin-embedded and sectioned at 10 μm thickness onto charged Superfrost Plus slides (LabServ, Thermo Fisher Scientific, Waltham, MA, USA). Following deparaffinisation with Xylene, the slides were rehydrated using a decreasing series of ethanol dilutions and subjected to heat-induced epitope retrieval. The sections were stained with primary antibodies (Appendix A) overnight at 4 °C, and the following day, they were exposed to enzyme-linked HRP secondary antibodies for 1 h at room temperature. A detailed IHC protocol is found in the Appendix A.

### 2.6. Statistical Analysis

For individual gene expression analysis, Gaussian distribution and homoscedasticity of data were tested using Shapiro–Wilks and Levene’s Test for Equality of Variances, respectively. Data with unequal variances were subjected to natural log transformation to enable parametric statistical testing. All normal or transformed data were analysed using a one-way analysis of variance (ANOVA) or Student’s t test when comparing two groups of one variable. One-way ANOVA tests were followed by a Bonferroni post hoc test (for univariate multiple comparisons), where applicable. All statistical analyses were performed using IBM SPSS Statistics Software (Version 26) or GraphPad Prism (Version 9), and the threshold for statistical significance was *p* < 0.05 unless otherwise stated. Data were graphed using GraphPad Prism (Version 9) and expressed as mean ± SEM unless otherwise stated.

For principal component analysis (PCA) and unsupervised hierarchical clustering, relative gene expression values for all samples were uploaded to ClustVis [34]. Volcano plots were produced using VolcaNoseR [35], where thresholds were defined for the *x*-axis as log fold change of ≥1.5 and *y*-axis significance of *p* < 0.05. For Gene Set Enrichment Analyses, the Broad Institute open-access software was used [36,37]. Pathways were defined using gene ontology terms curated in the mouse genome database. Only pathways with a false discovery rate of >25% and *p* < 0.05 were considered significant and, therefore, included in analyses.

## 3. Results

### 3.1. Putative Precursor Lesion and a Late-Stage Mouse Ovarian Tumour Phenotype Are Genetically Linked

Previously, we and others reported that *Fancd2^−/−^* mice develop multiple benign and malignant ovarian cancer phenotypes [17,32,33], including cystadenomas, cystadenocarcinomas, tubulostromal adenoma and adenocarcinomas, based on classifications defined by Mohr 2001 [38]. Therefore, to select the tumour phenotype investigated in this study, haematoxylin and eosin (H&E) staining was initially performed to identify samples that presented similar histology [38] (data not shown). Then, to confirm the tumour phenotype, immunohistochemistry (IHC) (Appendix A) was utilised to qualitatively assess a panel of common ovarian cancer diagnostic markers, such as calretinin, E-cadherin, paired-box 8 (Pax8), antigen KI-67 (Ki-67) and tumour protein P53 (p53) (Appendix A), with expression summarised in Appendix A. Calretinin, E-cadherin and Pax8 are commonly used to establish the ovarian tumour phenotype, namely sex-cord stromal versus epithelial phenotypes, while Ki-67 and p53 are indicative of the proliferative index and tumour grade, respectively. Three-month *Fancd2^+/+^* and *Fancd2^−/−^* ovarian cell types were positive for their expected markers, such as calretinin in the ovarian stroma, E-cadherin in the OSE, and Ki-67 and p53 in the highly dynamic granulosa cells and limited staining for the other ovarian cell types. The 1-year *Fancd2^−/−^* tissue displayed low Ki-67 and diffuse p53 staining, indicating low proliferative index and wild-type p53 function, and has mixed epithelial and stromal characteristics, confirmed by the presence of both the stromal marker calretinin and epithelial markers E-cadherin and Pax8. Therefore, the IHC data and observed characteristic nodular structures, presenting with variously sized tubule structures lined with squamous to cuboidal epithelial cells, are consistent with the murine tubulostromal adenoma OC phenotype [38,39].

Once a tumour phenotype was established, a multiplex gene expression assay was used to determine a genetic link between the identified precursor lesion of the sex cords in ovaries of 3-month *Fancd2^−/−^* mice and the tubulostromal adenoma ovarian tumour phenotype in 1-year old *Fancd2^−/−^* mice. Since tumorigenesis in this model depends on the pregranulosa cells within the primordial and primary follicles retaining a naïve epithelial state, instead of a differentiated GC phenotype present in growing follicles, GCs from these follicular types were collected as controls from 3-month and 1-year *Fancd2^+/+^* mice (Figure 1). A representative example of LCM for mature GCs and sex cords is depicted in Appendix A.

The gene expression data from GCs of primordial and primary follicles and from the growing follicles of 3-month and 1-year *Fancd2*^+/+^ animals were combined into naïve and mature GC, respectively. As the data points within each of these groupings were contained within discrete areas after PCA (Appendix A), it was assumed there were few (if any) gene expression differences between these groupings. Moreover, when pairwise comparisons were made between these groups, there were no differentially regulated genes (Appendix A). Therefore, these phenotypes were subsequently combined into naïve GCs (from primary and primordial follicles) and mature GCs (from growing follicles of 3-month and 1-year mice) for all downstream analyses.

The PCA based on the first two principal components (equivalent to ~41.6% of the total variance; Figure 2A), as well as unsupervised hierarchical clustering of relative gene expression across all GeXP gene sets (Figure 2B,C) resulted in clustering with minimal misclassification. Firstly, the phenotypes studied were segregated into their respective genotypes (WT vs. KO) (Figure 2B). Importantly, the precursor lesion, i.e., sex cords, segregated halfway between the naïve GCs from *Fancd2^+/+^* mice and the late-stage tumour phenotype of the *Fancd2^−/−^* mice (Figure 2B,C).

To ensure that these results were not solely an effect of genotype, ovarian stromal cells were included in a PCA and hierarchical clustering with all other phenotypes (Appendix A). The stromal cell gene expression from the different genotypes overlapped and clustered intermediately between the sex-cord and late-stage tumour phenotype samples (Appendix A). This provided evidence that the segregation of the WT and KO samples and consequently, the precursor and tumour samples, was not solely due to the underlying genotype, but to the premature loss of the regulating germ cells within the tissue.

To examine more closely the specific gene expression changes occurring, multiple pairwise comparisons were made and represented using volcano plots (Figure 3) displaying genes with *p*-values of <0.05 and an expression fold change of ≥1.5. Consistent with the PCA and unsupervised hierarchical clustering analysis, sex cords and the late-stage phenotype resulted in the least number of significant gene expression differences, with only one gene differentially expressed, the proto-oncogene *Kras* (*p* = 0.002) (Figure 3E), which was upregulated in the adenoma compared to the sex cords. Similarly, sex cords and the late-stage tumour cells both had numerous genes upregulated when directly compared to naïve GCs (Figure 3A,B, respectively), of which about half were similarly upregulated (*Alk5, Emp1, Igf1, Kitl, Pcna, Tdp2, Tead1 and Trp53*). The gene expression differences were more diverse when the sex cords and adenomas were compared to mature GCs, represented by both upregulated and downregulated genes, with more differences between sex cords and mature GCs than between the adenoma and mature GCs (Figure 3C,D). However, similarities did exist between genes that were downregulated, such as *Brca1 and Amh*, and upregulated, including *Pgc1α1* (Figure 3C,D).

Of the genes that were differentially expressed, there were multiple genes related to folliculogenesis within the sex cords compared to both naïve (Figure 3A) and mature (Figure 3C) GCs, such as an increased expression of *Bmpr2*, *Alk5, Alk6* and *Lhr* relative to naïve GCs and a decreased expression of *Bmpr2, Amh, Fshr and Foxl2* compared to mature GCs. Unsurprisingly, there were genes downregulated in the sex cords and adenoma samples, when compared to normal (mature) GCs, that are associated with DNA repair mechanisms, such as *Brca1, Brca2* and *Xrcc2* (Figure 3C,D), relating to both the loss of *Fancd2* in the knock-out phenotypes and the progression to an abnormal phenotype.

### 3.2. Sex Cords Express Both Epithelial and Granulosa Cell Marker Genes

To test the hypothesis that sex cords develop epithelial versus granulosa cell properties, the relative gene expression levels of epithelial markers of OC tumorigenesis, such as *Pax8, Cdh1* (encodes E-cadherin), *Muc16* (encodes CA-125) and *Wfdc2* (encodes HE4), and of folliculogenesis genes primarily upregulated in GCs in response to gonadotropins (*Lhr*), steroids *(Ahr*)*,* TGF-β superfamily family members (*Alk5, Alk6* and *Bmpr2)* or canonical markers of GC function (*Foxl2* and *Amh*) (Figure 4) were assessed individually from the GeXP multiplexed data.

The expression levels of *Pax8*, *Cdh1* and *Muc16* (Figure 4A) trended to be higher in both the sex cords and adenoma samples, as compared to the naïve and mature GC controls. The expression of *Pax8* was significantly higher in sex cords versus mature GCs (*p* = 0.029), and *Cdh1* (*p* = 0.017) expression was significantly higher in the adenoma compared to all other phenotype groups. Furthermore, *Muc16* trended to be the most highly expressed in sex cords but was not statistically significant due to gene expression variation between samples. Additionally, *Muc16* was statistically more highly expressed in adenomas compared to the mature GCs (*p* = 0.002) but was not different to that in sex cords (*p* = 0.117), even though average gene expression trended higher in the sex cords. Lastly, *Wfdc2* (Figure 4A), the gene related to a common clinical secretory biomarker HE4, was lowly expressed across all samples.

The resultant data analyses of the folliculogenesis genes revealed that *Lhr* and *Ahr* mRNA in the sex cord and adenoma phenotype, and *Foxl2* and *Bmpr2* mRNA in tubulostromal adenomas, were similarly expressed to that in mature GCs, with levels significantly different to the naïve cells (Figure 4B). However, the *Foxl2, Bmpr2* and *Amh* expressions in sex cords were not significantly different to naïve GCs. Additionally, the *Alk5* and *Alk6* mRNA were significantly increased in mature granulosa cells in comparison to the other phenotypes, apart from *Alk5* also being highly expressed in tubulostromal adenomas. Moreover, the expression levels of these two growth factor receptor genes were similar in the sex cords and adenomas and higher than that in naïve GCs.

### 3.3. Functional Classes of Genes Enriched in Relevant Model Phenotypes

To investigate the potential implications of the differentially expressed genes between the phenotypes on important pathways, gene ontology (GO) analyses (FDR <25% and *p* < 0.05) of biological pathways (BPs), molecular functions (MFs) and cellular components (CCs) were performed. All pairwise comparisons were made on the cell groups collected (summarised in Appendix A), and neither naïve GCs nor adenomas, compared to all other phenotype groups, resulted in enriched pathways. However, all pairwise comparisons with mature GCs resulted in enriched pathways (Figure 5, columns 1–3). Numerous pathways were enriched in the sex cords when compared to naïve GCs (Figure 5, column 4), but none in comparison to the other cell types, except one GO term that was enriched when compared to the adenoma phenotype (Appendix A).

The top four enriched biological pathways in mature, compared to naïve GCs were, reproduction, growth, enzyme-linked receptor protein signalling and regulation of the cell cycle, which were all likely related to the functional role of the GC within the follicle (Figure 5, column 1). Furthermore, most pathways enriched in mature GCs, compared to the adenoma samples, were related to DNA, metabolism and cell cycle stability, exemplified by negative regulation of biosynthetic and nucleobase-containing compound metabolic processes, mitotic cell cycle and cell cycle-related processes and cellular response to DNA damage stimuli (Figure 5, column 3). Interestingly, some pathways were similarly enriched in mature GCs when compared to the sex cords (Figure 5, column 2) and adenomas (Figure 5, column 3), such as DNA metabolic processes and negative regulation of biosynthetic and nucleobase-containing compound metabolic processes. Moreover, the epithelial cell proliferation pathway was enriched in the sex cords, compared to naïve GC (Figure 5, column 4), supporting our hypothesis that these cells revert to their epithelial phenotype, which is diminished in the presence of a germ cell. Finally, consistent with the gene expression data analyses, only one pathway was enriched in sex cords compared to that in adenomas. This pathway was the biological pathway of regulation of cellular component movement; however, this was not statistically significant (*p* = 0.053), supporting our hypothesis that these phenotypes are closely related.

## 4. Discussion

For the first time, this study demonstrates a genetic relationship between the sex cords and a late-stage ovarian tumour in *Fancd2^−/−^* mice, providing robust evidence that this is a precursor lesion in this animal model. Previously, the association of the sex cords as precursor lesions was hypothesised based on our group’s extensive knowledge of follicular development and a qualitative immunohistochemical study [17]; thus, this study builds on the former, solely observational research. Additionally, the data presented herein provide an explanation for an additional pathway in which oocyte loss after birth from non-growing (i.e., primordial and primary) follicles, due to exposure to X-irradiation [40] or dimethylbenzanthrene [41], gene knock-outs [17,19,32,42,43,44], genetic disorders (e.g., Fanconi Anaemia) [45] and genetic mutations [46], irreversibly leads to the development of ovarian tumours.

This data supports a novel paradigm of ovarian tumour development. Previously, the genesis of tubulostromal adenomas in other ‘loss of oocyte’ models has been attributed to the loss of follicles and consequent ovarian stromal reorganisation resulting in the extensive invasion of the ovarian surface epithelium, which is subsequently the putative source of preneoplastic precursors in the ovary [47]. However, the sex cords containing epithelial-derived granulosa cells are present without such a transformation in the OSE and thus are the proposed pathway for epithelial phenotypes developing within the ovary proper. Importantly though, we are not proposing that these mice develop ovarian tumours in an identical manner to human disease, as in fact, tubulostromal adenomas are specific to mouse ovarian tumour development [39], nor that this novel paradigm negates other known OC origins. Rather, we suggest that this model supports a novel paradigm of tumorigenesis and an opportunity to repurpose a known animal model for EOC as a tool to study early tumorigenesis since a definitive precursor lesion has now been identified and validated. However, it is unclear whether the tubulostromal adenoma phenotype studied is an intermediary between the sex cord precursor and other malignant phenotypes that may develop in this model, such as tubulostromal adenocarcinomas. Thus, in the future, it would be of interest to further investigate the relationship between the newly confirmed sex cord precursor and an adenocarcinoma phenotype known to develop in the *Fancd2^−/−^* model.

Using a multiplexed gene expression approach, we demonstrated that both the sex cords (precursor) and adenoma (late-stage) were the most similar based on PCA and unsupervised hierarchical clustering. We also confirmed that the differences in tissues between *Fancd2^−/−^* and *Fancd2^+/+^* mice were not exclusively linked to genotype differences, as confirmed with the stromal comparison controls. Of further note, the sex cords were also closely related to naïve GCs in both the PCA and hierarchical clustering analysis, suggesting that these two phenotypes were genetically related, thus supporting the novel tumorigenesis paradigm in this model. Unique to hierarchical clustering, the sex cords and adenoma samples were more closely related to the mature, than to the naïve, GCs. This can be explained by the reduced transcriptional activity in general of naïve GCs under the paracrine control of the oocyte in a non-growing follicle compared to the other phenotypes studied. This is supported by the many pathways that were enriched in the sex cords versus the naïve GCs (Figure 5), indicating that they have unique developmental trajectories. Additionally, the gene ontology analysis supported the biological context of the gene expression data, since the top four enriched pathways in mature versus naïve GCs were associated with the developmental maturation of GCs, and many of the enriched pathways in mature GCs versus the sex cord and adenoma phenotypes were attributed to loss of normal cellular functions, presumably due to the early and progressive transformations that occur in these knock-out tissues. However, these outcomes of the analyses included some bias since the genes investigated were selected upon certain cellular processes known to be important for tumour formation and folliculogenesis. Nevertheless, there were no enriched pathways specifically related to folliculogenesis found in the comparison between the mature and/or naïve GCs and the sex cord and adenoma, suggesting that the transformed phenotypes similarly retained some functions common to GC development. Finally, there were many enriched pathways in the sex cords compared to naïve GCs, including epithelial cell proliferation. The other pathways enriched between these phenotypes were likely due to the upregulation of processes required for the transformation of sex cords into pathological phenotypes, such as the late-stage adenoma; however, more investigation is required.

To address our key hypothesis that after premature loss of the oocyte, the sex cords formed follow an epithelial trajectory, individual gene expression analysis of epithelial markers were included in the GeXP sets. These results demonstrated that sex cords and adenomas exhibited similar expression levels of the epithelial markers (Figure 4A) of OC that were analysed (*Pax8* and *Cdh1*) and trended to be increased in comparison to WT tissues. Interestingly, *Muc16* expression was generally higher in the sex cords compared to all other tissues analysed, but due to a high sample variance, this was not statistically significant. Previously, *Pax8, Cdh1* and *Muc16* were reported to be expressed in normal FTE, with only moderate expression in the OSE, which increased within cortical inclusion cysts and in high-grade serous OC [48,49]. However, our gene expression and immunohistochemistry results herein reveal that these epithelial markers were also expressed in the sex cords within the ovarian cortex. Since *Fancd2^−/−^* mice do not ovulate [17], the expression of these epithelial markers within the ovary itself cannot be attributed to the introduction of transformed oviductal epithelial cells or the OSE via inclusion cysts. This unexpected expression pattern relates to this model’s mode of tumour development and is likely a contributing factor to OC tumorigenesis. Therefore, because the sex cords express characteristic epithelial markers, this provides a potential explanation for how cells within the ovary proper may serve as the precursor lesion for EOC in this animal model. Furthermore, individual analyses of folliculogenesis markers revealed that the expression levels of *Lhr, Ahr, Alk5* and *Alk6* trended to be similar in mature granulosa cells, sex cords and adenomas. Whilst it is not unexpected that these folliculogenesis markers were not highly expressed in naïve GCs, as these are upregulated within the GC in response to oocyte-derived growth factors and gonadotrophins following the activation of follicular growth, it is interesting that they are upregulated in the sex cords. Whilst this study did not include an exhaustive list of folliculogenesis and epithelial gene markers, those that were investigated provided genetic evidence that the sex cords and adenomas express both characteristic epithelial and follicular gene biomarkers.

## 5. Conclusions

This study provides initial genetic evidence that the sex cord can be classified as a bona fide precursor to adenomas in *Fancd2^−/−^* mice. Even though the relationship was only studied between sex cords and tubulostromal adenomas, given that multiple tumour phenotypes develop in this EOC mouse model, it is not unlikely that multiple tumour phenotypes derive from this putative precursor. Furthermore, our gene expression results provide strong evidence that pregranulosa cells remain multipotent in nature and, due to their epithelial developmental origins, can revert to an epithelial phenotype if abandoned by their adjacent germ cell regulator and contribute to the development of ovarian tumours. Thus, we genetically characterised *Fancd2^−/−^* mice for the first time as a model of early ovarian tumorigenesis. As such, we propose this model can be used broadly in future studies to further investigate early tissue and systemic biomarkers of ovarian tumours and, potentially also, ovarian malignancies.

## Figures and Tables

**Figure 1 cancers-15-02595-f001:**
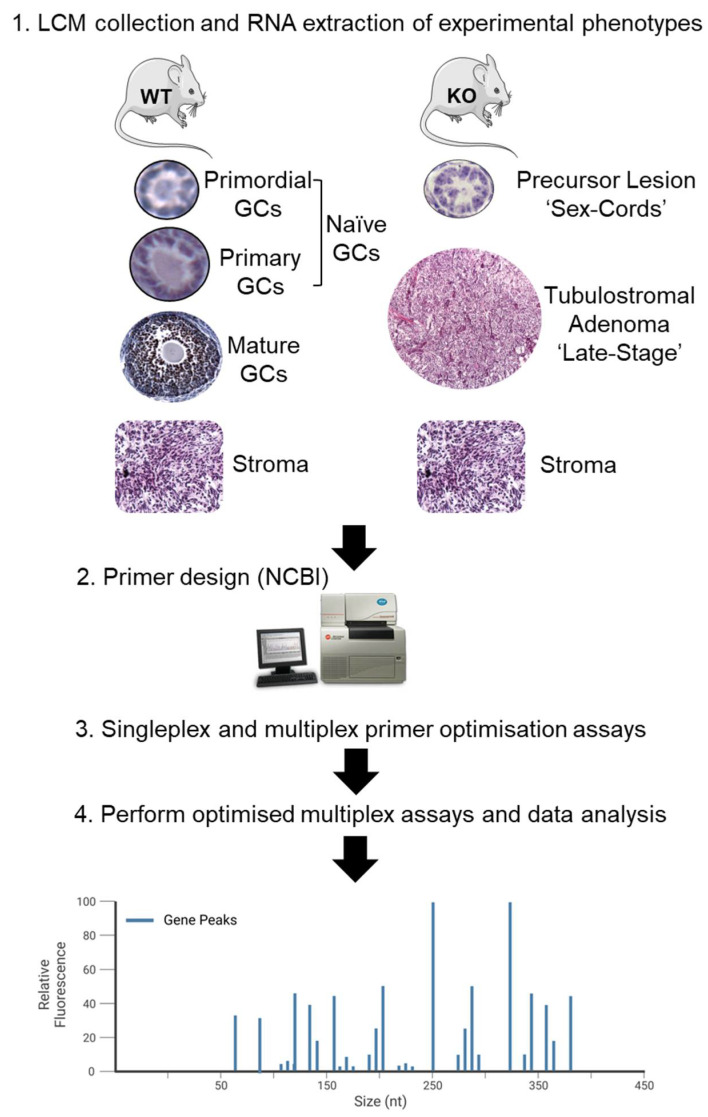
Schematic illustration showing the workflow for the analysis of relevant tissue phenotypes and controls using the Genome GeXP machine for multiplexed gene expression analysis. The precursor lesion and a late-stage mouse epithelial ovarian cancer phenotype were collected using laser capture microdissection from 3-month and 1-year animals, respectively, along with appropriate WT controls. In addition to the phenotypes of interest, ovarian stromal cells were collected from both 3-month WT and KO animals, to ensure any gene effects were due to the tissue phenotype, rather than genotype. GeXP primer sets were designed and tested, firstly in singleplex, and then in concert within three multiplex sets. LCM samples were then reverse-transcribed, and PCR amplified and optimised GeXP assays were performed. Adapted in part using Servier Medical Art and BioRender.com.

**Figure 2 cancers-15-02595-f002:**
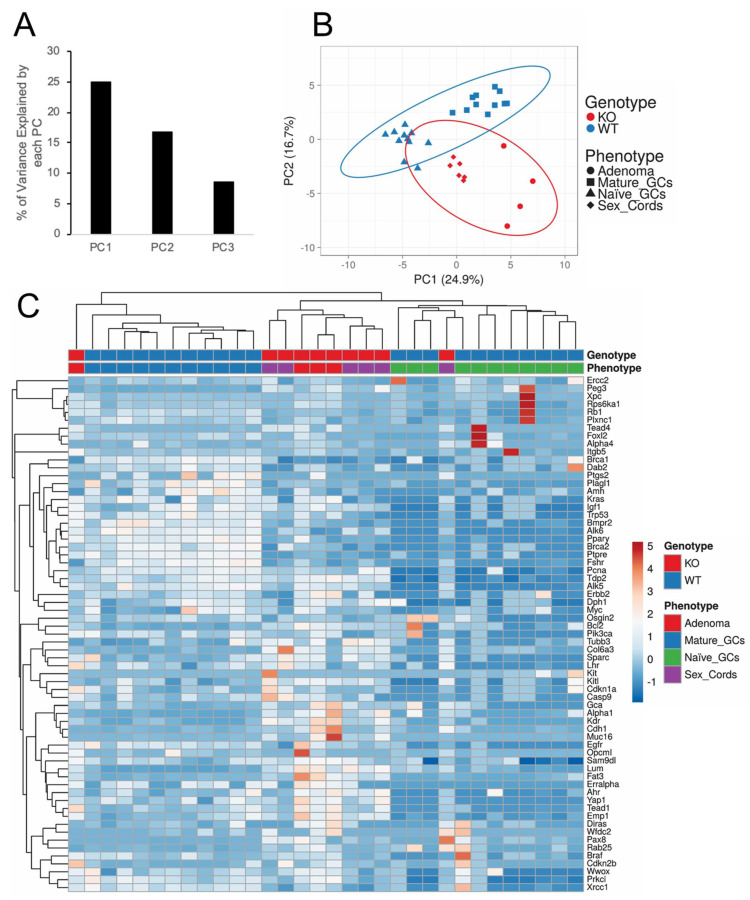
Unbiased data analysis of GeXP relative gene expression data depicting a close relationship between the sex cords and adenoma samples. (**A**) Bar graph displaying the per cent variance represented by the first three principal components. (**B**) *X*-axis and *Y*-axis showing principal component 1 and principal component 2, which explain 24.9% and 16.7% of the total variance, respectively. Data segregate based on genotype, WT vs. KO, with sex cords and adenomas being closely related. Sex cords are also closely related to naïve GCs. Prediction ellipses are such that with a probability of 0.95, a new observation from the same group will fall inside the ellipse. *N* = 33 data points. (**C**) Heat map showing the unsupervised hierarchical clustering with minimal misclassification. The data mirrors the results of the PCA plot. Rows are centred; unit variance scaling was applied to rows. Imputation was used for missing value estimation. Both rows and columns are clustered using correlation distance and average linkage. There are a total of 65 rows and 33 columns. PC = principal component.

**Figure 3 cancers-15-02595-f003:**
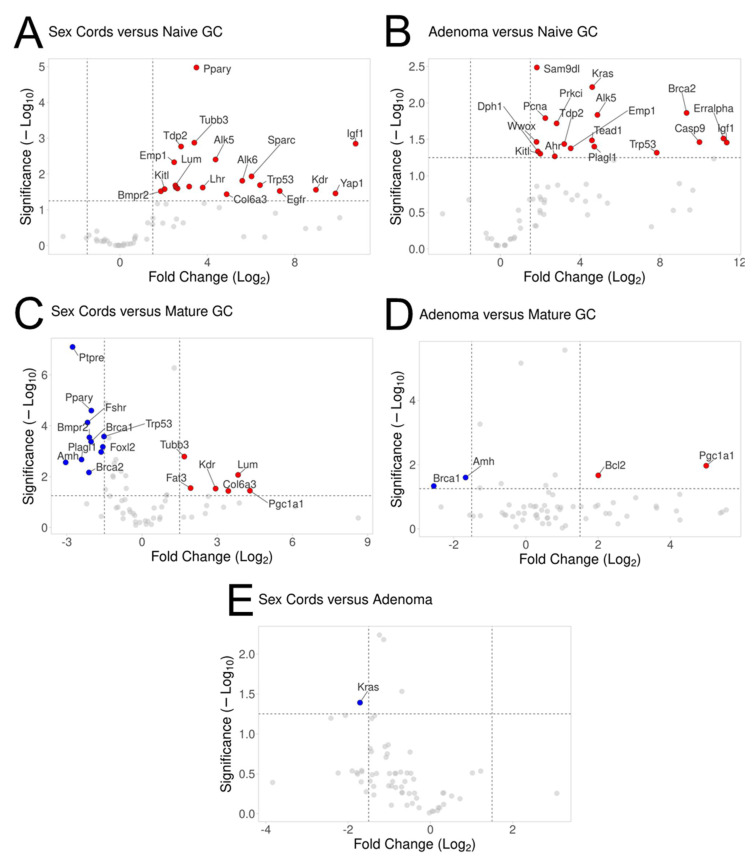
Volcano plots representing significant gene expression changes between all experimental sample pairwise comparisons. Naïve GCs and mature GCs versus (**A**,**C**) sex cords and (**B**,**D**) late-stage adenoma and (**E**) sex cords compared to the adenoma sample. Thresholds, represented as dashed lines, were set at a fold change > 1.5 and *p*–value *p* < 0.05 of relative gene expression data from all GeXP gene sets. Only genes above the designated thresholds are displayed on the graphs, with those in red representing a positive fold change (increased expression) and in blue representing a negative fold change (decreased expression) for the phenotype entitled second as compared to the first. Grey points represent genes below the threshold of significance, and a fold change is determined by the entitled A versus B comparison, i.e., A/B.

**Figure 4 cancers-15-02595-f004:**
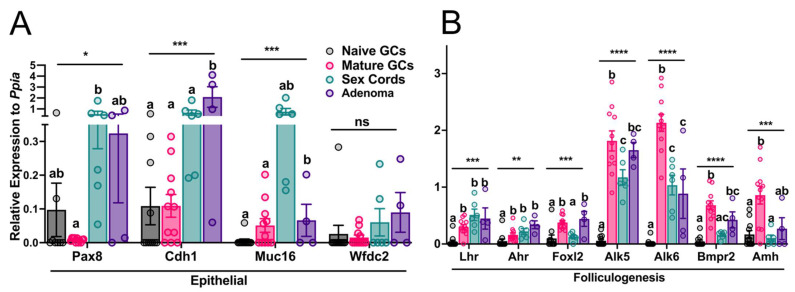
Individual gene expression results showing the (**A**) epithelial and (**B**) folliculogenesis markers included in all GeXP multiplex assays. All genes were analysed using one-way ANOVA. Data for all genes represent *n* = 12 for naïve GCs, *n* = 11 for mature GCs, *n* = 6 for sex cords and *n* = 4 for adenoma samples. Where a statistical difference was identified (denoted with an asterisk: * *p* < 0.05, ** *p* < 0.01, *** *p* < 0.001, **** *p* < 0.0001), a Bonferroni post hoc test for pairwise comparisons was performed in which groups not sharing a common letter were statistically different (*p* < 0.05). Relative expression is represented using the average fluorescence values across biological replicates and normalized to the endogenous control, *Ppia*.

**Figure 5 cancers-15-02595-f005:**
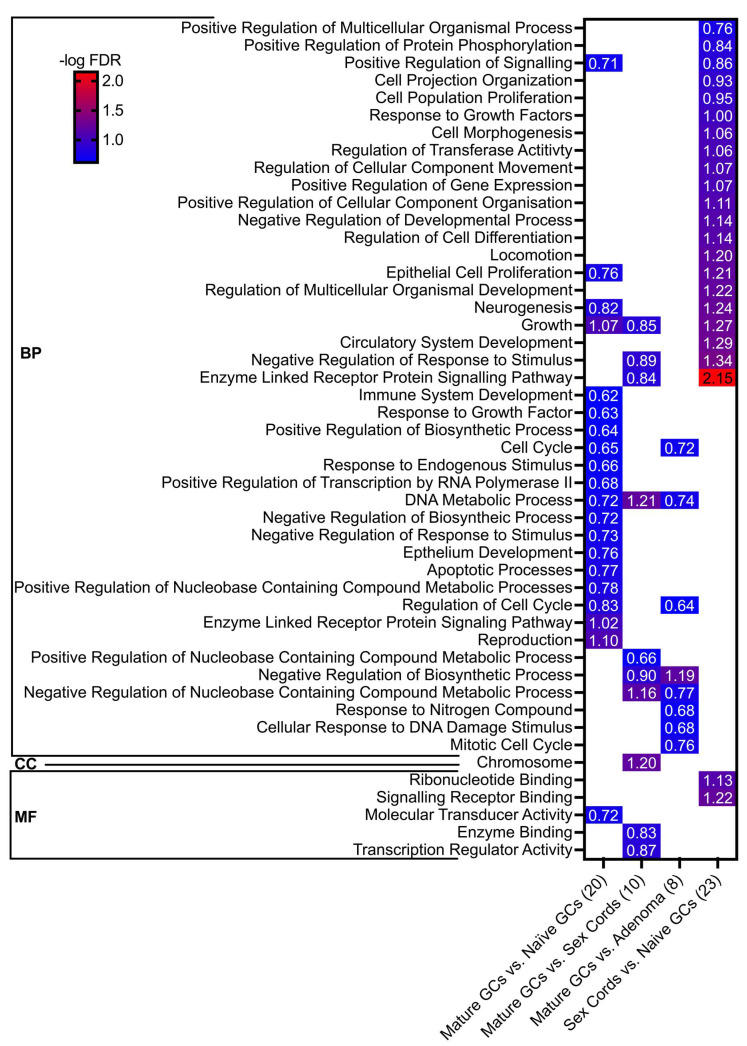
Functional enrichment analysis summary. Gene set enrichment analyses of the differentially expressed relative gene expression value. All pairwise comparisons were made, and only those that resulted in enriched pathways are displayed, with the number of pathways enriched contained in brackets after the comparison label. The threshold for significant gene ontology terms was assigned as FDR < 0.25 and *p* < 0.05, with red representing the highest, and blue the lowest, –log FDR values. Blank spaces represent no enrichment for that term in the comparison made. GO classifications are grouped into the biological process (BP), cellular component (CC) or molecular function (MF) terms. Versus (vs.) refers to pathways enriched in A as compared to (vs.) B.

## Data Availability

Data will be made available to any party upon request.

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
