# Peer review of "The Validation of a Precursor Lesion of Epithelial Ovarian Cancer in Fancd2-KO Mice"

_cancers, 2023, doi:10.3390/cancers15092595_

Round 1

Reviewer 1 Report

The manuscript is dealing with the ovarian cancer mouse model. The mouse model was developed earlier and is lacking Fancd2 DNA repair factor. Knockout mice were used for the experiments, and wild-type mice from the same cohort, potentially siblings, were used as controls.

The manuscript is well-written. However, the presentation of Figures needs to be improved, the novelty justified, and the conclusions sharpened. Moreover, the study is done using mainly RNA levels as a gene expression marker. The proteins were used for control the samples rather than to validate the new data obtained in the study.

1. If available in the literature or known based on the authors' observations, it would be interesting to include the following points to the results or discussion sections:

1a) Optional. What is the survival/mortality rate of Fancd2 mice? It is said that the mice are infertile, however, the experiments were performed using 12-month-old mice. What proportion of Fancd2-/- mice survive until 12 months of age? Is it similar or significantly different compared to Fancd2+/- and Fancd2+/+?

1b) Optional. What is the cancer rate in Fancd2+/- mice? Is it closer to WT, Fanckd2-/- knockouts, or somewhere in between?

2. Section 2.1 Mice. How were the mice genotyped? Kindly provide the information. Illustrate if possible (DNA sequencing, PCR, western blot validation?) There are various methods of genotyping used, and it might be of interest to readers regarding this particular publication or these mice in general.

3. Figure 2, especially 2A and 2C - the font is too small and not sharp enough for comfortable reading. Kindly fix the font and Figure resolution. Consider using the same font for similar elements on the same Figure. 

4. Optional. Figure 3. "v." probably means "vs"? "vs" is a common way to present "versus". It would improve the presentation if used "vs", "versus", or the meaning of "v." is explained in the Figure legend. Another Figure in the same paper is using "vs" instead of "v."

5. Figure 3. "E" appears twice, as the name of panel "E" and as an irrelevant letter on panel "C".  Kindly check.

6. Figure 3. The labels on the axes are too small and hardly readable. Consider fixing and making them consistent with other Figures. 

7. Figure 4. Kindly add to the Figure legend explanation of how the gene expression was measured here in addition to mentioning GeXP assay (e.g., RNA levels normalized to Ppia)

8. Figure 5A. Kindly use the same font for the letters. it seems that "Adeno" is using a different font than other labels. 

9. Figure 5 (B, C, D, E) - kindly use large font for labels and ensure the letters are sharp. It is hardly readable as it is now. Align "B" and "C". Use the same font for similar elements on the Figure.

10. The conclusion is too long and not sharp enough. It is clear that ovarian cancer was previously associated with this mouse model and thus it is not very novel study. It is also clear that the study is based on the RNA level and no validation with protein level was demonstrated. The statement that the protein expression was used is misleading as the protein expression was used to validate the samples being analyzed rather than validate the protein expression which potentially correlates with RNA levels. The conclusions could be sharper, organize as bullet points, and justify the novelty of the study.

Reviewer 2 Report

 This article focused on the expression of ovarian cancer (OC) related 90 genes comparing “Sex Cord”, “Adenoma”, “stroma” in Fancd2-/- mice and “Naïve GC”, “Mature GC”, “stroma” in WT mice, suggesting premature loss of oocyte leads “Sex Cord” to epithelial tumor trajectory. Although the authors regard “Sex Cord” as the precursor of adenoma based on this research using only this Fancd2-/- mice and just biased OC-related 90 genes bioinformatics analyses, it might be worth investigating this theory for EOC carcinogenesis mechanism. This manuscript is generally well-written, however, I think the article might be acceptable after some corrections have been done.

Major points:

 1. As “Simple Summary” is not reflected from “Abstract” or “Conclusions” very much, the authors should revise it.

 2. Although the authors described that Fancd2-/- mice develop multiple ovarian tumor phenotypes as they age, the authors investigated the adenoma samples (n=4) only, not including OC. Also, as there is no mention of the detailed histologic characteristics of the adenoma samples (n=4), it is necessary to show all the adenoma histology. In addition, please explain why the number of adenoma samples was set as 4, as it is ideal that the adenoma samples (n=4) would be more. (Figure 4, Line 348)

3. Material and methods, Lines 124-126; Please explain in more detail the rationale for using 3-month-old mice as controls.

4. Discussion; As the authors think the mechanism of EOC carcinogenesis in Fancd2-/- mice is related to “loss of oocytes”, are there any previous reports that show the relationship between infertile knockout mouse models and EOC development to support the authors’ hypothesis?

Minor points:

1. Page 1, Lines 11-12; “demonstrate” might be changed to “demonstrated”.

2. Page 1, Line 16; “hampers” might be changed with “hamper”.

3. Page 1, Line 42; “repair it is” might be changed with “repair. It is”.

4. Page 3, Line 132; miRNeasy is not an ideal kit to extract total RNA.

5. Page 10, Lines 342-344; According to Figure 4A, Muc16 was more highly expressed in sex cords, not in adenomas.

6. Page 13, Lines 436-437; “are formed” might be changed with “formed”.

7. Page 13, Line 462; “investigated” might be changed with “investigation”.

8. Figure 1, Line 231; “laser-capture” should be replaced with “laser capture”.

9. Figure 2B and 2C; “Stroma” could be deleted in the Figure, as “Stroma” samples are not shown in the Figures. In the legend, is principal component 1 “24.9%”, not “29.4%”? (Line 257)

10. Figure 3C; I think “E” should be deleted.

11. Figure 5B, 5C, and 5E; The authors should describe the complete name of the pathways name, not described as “...".

12. Figure 5C; What does the green bar represent? Please explain it in the legend.

13. Figure 5B-E; As it is difficult to distinguish between orange and yellow bars, please change these colors.

14. Figure S1; “Ki67” should be replaced with ”Ki-67”. In addition, please add each H&E staining in this Figure.

15. Figure S4; The dots in volcano plats are too thin to recognize.

16. Table S5; “Ki67” should be replaced with ”Ki-67”.

17. Table S5; “P53” should be replaced with “p53”.

Reviewer 3 Report

The manuscript entitled, “The Validation of a Precursor Lesion of Epithelial Ovarian Cancer in Fancd2-KO Mice” is worth acceptance for publication in Cancers.

The research and results were well established and I have nothing to point out, however, I want to ask why the authors did not include ovarian cancer in Fancd2-/- mice which can be obtained if they breed Fancd2-/- mice longer.

I thought the similarity or somehow the difference between sex-cord cells and their ovarian cancer would give more impact on this study.

Round 2

Reviewer 1 Report

The authors used the opportunity to revise the manuscript based on the Reviewers' and Editor's suggestions. The presentation has improved and the questions raised during the first round of revision have been addressed.